# Lockdowns, lethality, and laissez-faire politics. Public discourses on political authorities in high-trust countries during the COVID-19 pandemic

**Sara Grøn Perlstein**[¤], **Marc Verboord***

Erasmus School of History, Culture and Communication, Erasmus University Rotterdam, Rotterdam, The Netherlands

¤ Current address: Institute of Security and Global Affairs, Faculty of Governance and Global Affairs, Leiden University, The Hague, The Netherlands
* verboord@eshcc.eur.nl

**Data Availability Statement:** All raw data is available from figshare (https://figshare.com/projects/Lockdowns_Lethality_and_Laissez-Faire_

## Abstract

This study looks at population response to government containment strategies during initial stages of the COVID-19 pandemic in four high-trust Northern European countries–Denmark, Germany, the Netherlands, and Sweden–with special emphasis on expressions of governmental trust. Sentiment analysis and topic modeling analysis were performed using Twitter data from three phases during the initial European lockdown, and results were compared over time and between countries. Findings show that, in line with existing theory, assertive crisis responses and proactive communication were generally well-received, whereas tentative crisis responses or indications by the authorities that the crisis was manageable were generally met with suspicion. In addition, while government support was high in all countries during the height of the crisis, messages critical of the government as well as conspiracy theories were nevertheless widely circulated. Importantly, countries with the least assertive strategies, rather than clear negative responses, saw heightened polarization of sentiment in the population. Furthermore, in the case of Sweden, a laissez-faire strategy was generally accepted by the population, despite strong criticism from other countries, until mortality rates started to rise. Possible explanations for these findings are discussed with an emphasis of prior trust as a potential explanatory factor. Future research should seek to replicate these findings in other countries with different levels of prior governmental trust or with a different severity of the COVID-19 outbreak than the countries in this study as well as triangulate the findings of this study using alternative methods.

## Introduction

When COVID-19 hit Europe in early 2020, political leaders were faced with a challenge without political precedence [1]. The novel coronavirus was spreading with a rapidity and on a scale not seen since the 1918 Spanish Flu [2, 3]. Its lethality was yet unknown, as was

Politics/95909). All processed data may be found in the Supporting Information files.

**Funding:** The authors received no specific funding for this work.

**Competing interests:** The authors have declared that no competing interests exist.

its mode of infection [4, 5]. There were no known cures, treatment options, or prophylaxes. In short, European societies were faced with major impending social disruption. If political leaders put their countries into quarantine, numerous bankruptcies, mass unemployment, and economic recession loomed on the horizon, all without any guarantee of effectiveness [6–10]; if they did nothing, millions of people could die or suffer long-term injury. With no manual on how to maneuver the pandemic, individual countries were forced to decide the best approach for themselves; in consequence, 2020 saw numerous examples of countries, which, while historically, culturally, and socioeconomically similar, adopted markedly different strategies [11].

Previous epidemics have shown that population compliance with government restrictions can make or break outbreak containment efforts. We also know that one of the major determinants in how well citizens comply with government recommendations is trust [12–15]. So how can we understand the different trajectories of countries which are generally considered "high-trust" countries [16, 17]? While existing studies have exemplified the potential of social media data to capture public sentiments and concerns [18, 19], cross-national studies are scarcer. This paper examines how citizens in four high-trust countries–Denmark, Germany, the Netherlands, Sweden–addressed authorities on Twitter during the COVID-19 pandemic. Thus, the aim of the paper is to analyze *the discourses surrounding governmental authorities during the initial stages of the COVID-19 pandemic in four Northern European countries with very different outbreak containment strategies and scrutinize these discourses for notions of trust.* The study bases itself on Twitter data from three phases during the initial outbreak, which mark the initial presence of the virus ((from the day of the first Covid-19 case (respective to each country) until March 10, 2020), the beginning of the pan-European lockdown (March 11–25, 2020), and after the lockdown had been in place for a month (April 11–25, 2020). These three phases, then, should be informative with regards to public response to political leaders as the pandemic threat became visible, as the political leaders responded (or did not respond) to the threat, and when the consequences of the political response started to become apparent and/or fatigue with the restrictions set in. The analysis is performed by employing sentiment analysis and topic modeling analysis.

Population sentiment toward political authorities may be difficult to measure in crisis situations, either because the urgency of the situation makes surveys or interviews less feasible, or–such as in the case of the COVID-19 pandemic–quickly outdated. By the time a survey has been constructed and distributed, people's sentiments may have changed [13]. By performing sentiment analysis on Twitter data, it is possible to identify developments in sentiment–that is, the positive or negative valence of attitude–at very specific points in time. In the same way, topic modeling analysis makes it possible to identify which topics–and thus, which general discourses–are prominent on Twitter in the wake of specific political events [20].

We illustrate how big data analysis can contribute to facing societal challenges in a time of crisis. The article will inform the field of disaster research on how high-trust countries respond to different outbreak containment strategies. The findings will lay the groundwork for further studies investigating public response to government regulations during the COVID-19 pandemic, and they may inform research into public compliance both in relation to the COVID-19 pandemic specifically and to epidemics in general. In addition, the findings may assist policymakers in better anticipating and preempting public backlash to certain strategies in the future. These types of investigations are increasingly important, as the ever-increasing interconnectedness of the global community makes disaster events like the COVID-19 pandemic likely to occur at a higher frequency than has historically been the case [3, 21–24].

## Theoretical background

### Disaster sociology and the role of authorities during disasters

Within the context of large disasters, the importance of population compliance with government regulation is difficult to overstate. According to contemporary disaster research, the main characteristic of a disaster is the social disruption it causes: a level 12 tsunami hitting an unpopulated shore is unlikely to be designated a disaster, as is the seasonal influenza, which, despite an annual global death count of around 650,000, thus surpassing most singular catastrophic events, nevertheless causes no significant disruption to social macrosystems [25, 26]. And, while there is no denying that external instigators of disasters may cause considerable damage, this holds no less true of the resulting social disruption, as uncertainty as to how to navigate the changed circumstances causes a coordination deficit in society, which may prompt people to act in unpredictable ways [26–28].

Despite populations generally being willing to make sacrifices to navigate safely through a crisis, this does not necessarily imply they adhere to the recommendations made by the political authorities in charge. Non-compliance may occur for a variety of reasons, whether through conflicting social obligations, opportunist reasoning, or simply through distrust toward authorities–or even anti-authoritarian sentiments [29]. The issue of trust, in particular, becomes salient when disaster strikes. While political authorities are able to base their judgments of crisis situations on vast knowledge resources, whether through expertise or access to privileged information, citizens are limited, both in their access to knowledge resources, and in disposable time to inspect such resources. This leaves citizens at an information deficit, ultimately forcing them to base their judgments of the situation on their trust in the political authorities providing them with trustworthy information rather than on the information itself (Tampere et al., 2016, pp. 418f). For example, in a 2011 survey study based on the European Values Study, it was found that individuals with a low level of political trust are much more accepting of law non-compliance than those with high levels of political trust [17]. In a US study on people's hypothetical acceptance of mandatory vaccinations and quarantine in the event of a smallpox bioterror attack, it was found opponents of such measures were deeply distrustful of the government, expecting power abuses, discrimination, and general unpreparedness [14]. It is thus imperative for both social scientists and policy makers to take an interest in the thoughts and feelings prevalent in the population–especially, when these thoughts and feelings are an expression of varying levels of trust in political authorities.

According to Boin et al. [30], three factors determine the credibility of authorities during a crisis: prior trust, the initial response, and the timing of messages. Trust in authorities prior to a disaster is a major determinant in how initial crisis communication is received in the population since it forms the baseline of trust. The initial response is important in that it signals the capacity and willingness of authorities to handle the situation (see also [31]). During times of crisis, the population expects political authorities to take a dominant and active role in crisis management and to give guidance and direction [32]. The population may even be more forgiving of an exaggerated strategy than an insufficient one [30]. At the same time, if authorities downplay the gravity of a crisis or overplay their ability to solve the crisis, this type of rhetoric may initially be received positively, but could ultimately result in failed expectations [30] leading to frustration and disorientation in the short term and potential generalized distrust in the long term [30, 32, 33]. Furthermore, a lack of guidance and direction may lead citizens to seek out information from other sources, which are often faster and more comprehensive–especially in the case of social media–but less reliable. Citizens turning to alternative information sources could thus increase non-compliance [32, 34–38]. Lastly, the timing of messages is of considerable importance. Overdue crisis information may lead to accusations of "covering up

the truth", or it may simply cause authorities to lose momentum (see also [39, 40]). Rather than softening the blow of crisis communication through unfounded positivity or the withholding of information, research indicates that openness about the risk surprisingly helps to reduce perceived risk in the population [39, 41, 42]. In addition, giving the population concrete directions on how to stay safe may limit the negative impact of the risk communication and help people feel more in control of the situation [41, 42].

In summary, research shows that political authorities managing a crisis should be transparent surrounding risk, take a dominant and active role in crisis management, and give concrete directions to give citizens a sense of control. Should the authorities fail to do this, they run the risk of the population developing distrust toward them and seeking out alternative information sources, which may lead to lower compliance with government measures and behavioral recommendations.

## Social media data and natural language processing in disaster research

Social media have gained considerable popularity as a measure of public sentiment in recent years. While mass media like newspapers and news broadcasts have traditionally been used as indicators of prominent public discourse, they also have clear agenda-setting and gatekeeping functions; mass media-based findings on public discourses may thus indicate what news outlets consider important to the population rather than what the population itself considers important [43, 44]. Compared to the top-down communication of mass media, the direct bottom-up nature of social media allows researchers to study political information and discourses outside of institutional frameworks [45–48].

Social media research also provides improved research opportunities for disaster and health researchers. One of the main challenges of disaster research concerns restrictions in both timing and access: disaster events tend to happen suddenly and rapidly, limiting the amount of time researchers have to plan out their studies and gather data. And even when the researchers are onsite, the *access* to implicated individuals is often restricted and may change suddenly [49]. Social media posts, on the other hand, are often posted, not only immediately in the wake of disaster events, but as responses to the events, and their accessibility, as a rule, hinges on neither the geographical location of the researcher nor the date at which the data is collected [50–53]. As such, researchers may gain a unique insight into people's thoughts and feelings at the time of occurrence of a disaster event, even when the data is gathered sometime in the future [54, 55].

Also in the domain of health have research tweets been used by epidemiologists to track the spread of diseases such as flu and dengue fever [20]. In recent years, opinion mining techniques in social research has provided researchers with even more opportunities to utilize social media. Natural Language Processing techniques (henceforth: NLP techniques) allow researchers to extract information about the thoughts and feelings present in a text corpus, or about social and semantic networks. For example, Pruss et al. [56] applied topic modelling to gain insights in how the Zika disease was discussed during the outbreak in the Americas in 2015. Insights from such work are similar to those traditionally found in qualitative social research, but the large quantities of data that NLP techniques can process lend to the method a higher level of generalizability as well as allowing for a more quantitative approach.

## Outbreak containment strategies during the COVID-19 pandemic in Northern Europe

On March 11th, 2020, COVID-19 was declared a pandemic by the WHO. Due to the unfamiliarity of the situation, the four Northern European countries Denmark, Germany, the

Netherlands, and Sweden, despite being culturally and sociostructurally similar, were seen implementing highly divergent strategies. Statistical data consistently shows these four countries to have a high level of population trust in their political authorities [17, 57–62]; as such, these four countries provide a valuable case study of the effects of different containment strategies on public opinion in high-trust countries. Below we summarize their diverging strategies (for details see *S1 Appendix: Timelines (by country)*).

The Danish strategy was by far the most assertive strategy of the four: it introduced a total lockdown only a day after the WHO classified Covid-19 as a pandemic (March 12[th]). The goal of the containment strategy was to limit the spread until the development of a vaccine or more effective treatment options than were available at the time [63]. During the lockdown, the government enjoyed support both from a nigh united parliament and from the population at large [64, 65]. Overall, Denmark managed to contain the virus throughout with intensive care units running below capacity throughout [66].

Although Germany was one of the first European countries to be hit by the COVID-19 outbreak [67, 68], Germany only began to introduce containment measures after the WHO declared COVID-19 a global health pandemic on March 11[th] [69]. From this point onwards, however, the German strategy quickly evolved in a similar direction to the Danish strategy. Germany has the highest number of acute care beds per 1000 people in Europe and performed extensive testing, and as a result, saw low fatality rates throughout the period of this study [70, 71].

The Dutch government initially dubbed the Dutch strategy "intelligent lockdown", aiming to flatten the curve of infection, while at the same time allowing for herd immunity to take hold [72]. In concrete terms, this mean that the Netherlands introduced outbreak containment measures in several steps and only when deemed necessary [73]. Between March 15[th] and March 23[rd], the Dutch strategy became stricter, changing from a model akin to the Swedish model to becoming more similar to the strategies of Denmark and Germany [74, 75]. Nevertheless, the Netherlands had a high ratio of infected citizens and at one point had to send COVID-19 patients to Germany for treatment due to insufficient intensive care equipment [76, 77].

The Swedish containment strategy was the least restrictive of the four countries, emphasizing voluntarity and maintaining individual responsibility as a cornerstone of limiting the spread [78, 79]. Rather than attempting to stop the spread of the virus completely, the Swedish strategy aimed to contain the negative consequences of the outbreak with a special emphasis on keeping society up and running, and, as such, Sweden had no lockdown comparable to those of the other countries [79]. Sweden saw high infection rates with retirement homes in the Stockholm area being hit particularly hard, although state epidemiologist and head of the COVID-19 outbreak containment strategy, Anders Tegnell, generally maintained that the Swedish strategy was more sustainable in the long run, especially if a second wave of infection should occur [80, 81].

## Methodology

This study maps discourse on Twitter related to governmental COVID-19 strategies in the four Northern-European high-trust countries of Denmark, Germany, the Netherlands and Sweden during the first wave of COVID-19. For the purposes of this study, the first wave is defined as the period between January 27 2020, marking the date of occurrence of the first case in Germany, and April 25 2020, a month and a half after the WHO declaration of COVID-19 as a pandemic and the point. Within the first wave, three phases are analyzed, describable as 1) the first occurrences of cases of COVID-19 in the countries investigated, 2) the initial weeks of lockdowns, and 3) the period, during which the countries investigated began to lift restrictions (see section *Data collection* for additional details). Twitter discourse is mapped by way of two

natural language processing (NLP) techniques: sentiment analysis and topic modeling analysis. First, general Twitter sentiment was measured by conducting sentiment analysis on Tweets from each phase within each individual country. Second, topic modeling analysis was conducted to inform the sentiment analysis by identifying prevalent Twitter discourses within each country during each of the three phases.

The use of Twitter data for research purposes is generally considered to be ethical practice due to the public nature of Twitter as a platform. Furthermore, as our method of data retrieval does not include metadata, users may not be personally identified based on the contents of the datasets analyzed in this study. To provide an extra layer of protection, usernames have been removed from the datasets published together with this article.

## Data collection

Data was collected from the social media platform Twitter. The ethics of treating social media as public spaces has long been debated in the scientific community [82–84], and, while the distinction between public and private space is a lot less clearly defined online than in offline settings, there is general consensus among social media researchers that Twitter users do perceive Twitter as an unequivocally public space [85–87]; for this reason, informed consent was not obtained for this study. Nevertheless, in order to provide some level of protection of the Twitter users whose tweets were included in this study, usernames have been removed from the data made available in the supporting information for this study. The Twitter API limits the retrieval of tweets to tweets from the past week; in order to circumvent this limitation, we used the software GetOldTweets by Jefferson Henrique [88], which has gained some prominence among social media researchers in recent years, as it permits the collection of up to 100% of all available tweets under the search terms entered [85, 89–92]. As this method of data collection does not include any metadata, no personal data was collected within the context of this study. Although we aimed for a sample size of 100,000 per data set, this number was only actually achieved for the base sets (see below) of Germany, the Netherlands, and Sweden. As such, the data sets in the study, for the most part, constituted the entirety of the population of Twitter posts in the respective country during the respective period.

It must be taken into consideration that sentiment on Twitter may differ between countries, not only due to differences in culture, but due to differences in language use as well–or, indeed, as a function of the dictionaries used (while these were generally translated on the basis of the English dictionary, many sentiment scores cannot be transferred directly, and, as such, the dictionaries may differ considerably). A similar dislike for the sitting government may be expressed in one language—or even one geographical area—as "The government is doing terribly", but as "The government does not live up to expectations" in the next. For this reason, a base set for each country of up to 100,000 tweets from 2019 was collected to serve as a basis for the identification of changes in sentiment. The main data was collected within three phases during the initial outbreak, outlined in Table 1 below.

**Table 1. Data collection phases.**

|  | Start | End |
|---|---|---|
| **Phase 1** | January 27th (DE) / February 27th (DK, NL, SE[1]) | March 10th |
| **Phase 2** | March 11th | March 25th |
| **Phase 3** | April 11th | April 25th |

[1] While Sweden did see a case on January 31st already, this case did not lead to further contagion. Instead, the Covid-19 outbreak only began in Sweden when the illness was brought there by other travelers on February 27th.

**Table 2. Conversion of non-English letters.**

| Original letter | æ | ø | å | ä | ö | ü | ï | ë |
|---|---|---|---|---|---|---|---|---|
| Converted format | ae | oe | aa | ae | oe | ue | i | e |

The first phase marks the initial awareness of and reaction to COVID-19 as a potential threat. The World Health Organization's reclassification of Covid-19 as a pandemic on March 11th [93] marks the beginning of Phase 2, which was set to 2 weeks' duration in order to capture public sentiment and discourse in relation to the initial lockdown. Finally, Phase 3 begins on April 11th—a month after the pandemic reclassification—and is set to 2 weeks' duration, covering a time period, in which containment measures had been implemented for a month.

Search terms for the base set included commonly used terms to refer to the government and head of state. In the search terms for phases 1, 2, and 3, commonly used terms for COVID-19 were added. While Twitter does allow the user to search for geographically specific tweets, unfortunately this function is not only unavailable in the GetOldTweets software but, as only around 2% of tweets contain geographical information, choosing tweets based on geolocation also tends to severely limit the sample size [94]. As the majority of speakers within the respective language spheres of the four countries analyzed in this study belong to the countries in question, we operate under the assumption that our findings are applicable to the analyzed countries, even if there is a minority presence of speakers from other countries; nevertheless, in order retrieve as many relevant tweets as possible while excluding irrelevant ones, we have based our search terms on keywords specific to our analyzed countries while excluding keywords specific to countries not part of our analysis. For example, in the German tweets, one of the following terms had to be included in the tweet for the tweet to be retrieved: Merkel, Bundeskanzlerin, Kanzlerin, Regierung, or @AngelaMerkelCDU, whereas tweets including the terms referring to Austria or Switzerland (Österreich, Schweiz, Bundeskanzler, Bundespräsidentin, etc.–see *S2 Appendix: Twitter search terms* for the full inclusions and exclusions).

No cleaning was performed prior to the sentiment analysis, although due to a program error a few results could not be processed and were subsequently removed from the raw results before converting into percentage tables.

Cleaning prior to the topic modeling analysis involved two steps. First, all non-English letters had to be converted to English letters, as Context is not able to process non-English letters. An overview of this conversion can be found in Table 2 below.

The second step involved removing repetitive patterns of phrasing or hashtags by the same user, for example if a single user begins all their tweets with #COVID #Government #Policy #BreakingNews, as these strings otherwise take up several of the topics in the topic modeling analysis on their own. It was taken care to avoid performing this cleaning in the case of retweets.

## Method of analysis

In this study, the NLP techniques of dictionary-based sentiment analysis, using the software SentiStrength, and topic modeling, using the software ConText, were used [95, 96]. The combination of sentiment analysis and topic modeling allows us to obtain a nuanced picture on Twitter discourse, including both Twitter users' emotional state and the topics being discussed (for previous uses of the combination of these methods, see [19, 97–100].

Sentiment analysis detects the overall sentiment of a text corpus by identifying words holding positive or negative valence, e.g. *good* or *terrific* (positive), and *bad* or *disgusting* (negative) (Liu, 2012). The sentiment analysis software scans the text corpus and scores individual texts

(read: tweets) based on the presence of sentiment words as well as a limited number of grammatical modifiers. If, for instance, a sentiment word is preceded by a negating word, e.g. *not* or *never*, then the valence of the sentiment word is reversed: the sentence "Merkel is a competent leader" would be scored as positive (= "Merkel [0] is[0] a[0] **competent[1]** leader[0]"), but the sentence "Merkel is not a competent leader" would be scored as negative ("Merkel[0] is[0] **not** a[0] **competent[−1]** leader[0]"). In addition, some words may modify the strength of sentiment words, e.g. "Merkel[0] is[0] an[0] **extremely[+2] incompetent[−1]** leader[0]" = [0;-3]. A tweet may have a sentiment of [0;0], i.e. no sentiment, for example "Merkel[0] is[0] holding[0] a[0] speech[0] now[0]"; this does not exclude the tweet from the analysis. An average sentiment of 1 thus does not reveal if the data set consists mainly of tweets with a valence of 1 or mainly of no-sentiment tweets with a minority of tweets with a very strong valence (hence the breakdowns provided in Table 4 and, in greater detail, in *S5 Appendix: Sentiment analysis results*). The detection of sentiment words is conducted based on a sentiment lexicon, which lists all sentiment words of a language as well as their valence. Dutch and German sentiment dictionaries already exist, but for the purposes of this article, Danish and Swedish sentiment dictionaries were created (see *S3 Appendix: Sentiment dictionaries (DK and SE)*). Testing was done by manually inspecting a subsample of 100 tweets from the main sample and adjusting the dictionaries until the analysis was satisfactory. The results of the tests can be found in Table 3 below.

On average, the sentiment dictionaries have an identification rate of 69.5%. Although the score will likely be lower when applied to a new data set, this result is sufficient to identify the general sentiment of a data set. SentiStrength uses a scale from 0 to 4 for positive sentiment and 0 to -4 for negative sentiment, 0 being the absence of sentiment and 4/-4 being the strongest possible sentiment. The output consists of an overview of the frequencies of different sentiment score combinations as well as the average positive sentiment (henceforth: APS) and the average positive sentiment (henceforth: APS) for the entire text corpus.

Topic modeling analysis is a machine learning technique, which identifies clusters of commonly co-occurring words in a text corpus [101–103]. The researcher sets the number of topics K expected to be in the text corpus and the number of words-per-topic desired *a* and receives K lists of *a* words. Based on each list, referred to as a *topic*, the researcher then identifies the common theme of the words, thereby uncovering the underlying discourses of the text corpus. The words of each topic are sorted according to fit (best fit first), and each topic is given a weight, which denotes the average fit of the overall topic to the text corpus [96]. As

**Table 3. Pre- and post-test scores for dictionary test sets.**

|  |  | Correct (max. 1pt. deviation from human assessment) | Partially correct (2pt. deviation from human assessment) | Incorrect[*] |
|---|---|---|---|---|
| DK | Pre-test | 46 | 46 | 8 |
|  | **Post-test** | **70** | **26** | **4** |
| DE | Pre-test | 58 | 37 | 5 |
|  | **Post-test** | **73** | **25** | **2** |
| NL | Pre-test | 47 | 42 | 11 |
|  | **Post-test** | **57** | **35** | **8** |
| SE | Pre-test | 67 | 27 | 6 |
|  | **Post-test** | **78** | **20** | **2** |

[*]A score is deemed incorrect, either if it indicates sentiment where there is none, indicates no sentiment where there is sentiment, or if the strength of the sentiment indicated differs with more than one point compared to the human estimation (e.g. if SentiStrength indicates a positive sentiment of 2, but the human estimation is 4). If one score indicates a very strong sentiment where there is none, the tweet is added to the third column (incorrect).

**Table 4. Sentiment analysis results in detail.**

| Denmark | | | | | | | | | | |
|---|---|---|---|---|---|---|---|---|---|---|
| | Avg pos | Avg neg | No sent | Strong sent | Strong pos sent | Strong neg sent | Pos sent only | Neg sent only | Pos > neg | Neg > pos |
| **Base** | 0.603 | -0.805 | 25.2 | 6.15 | 2.29 | 4.0 | 16.31 | 31.75 | 22.84 | 37.69 |
| **Phase 1** | 0.513* | -0.997*** | 19.18 | 7.23 | 2.2 | 5.03 | 8.8 | 44.02 | 15.72 | 49.36 |
| **Phase 2** | 0.650****˚ | -0.914*** | 19.14 | 5.75 | 2.33 | 3.58 | 14.59 | 33.54 | 21.79 | 41.06 |
| **Phase 3** | 0.618 | -0.905*** | 19.5 | 6.08 | 1.7 | 4.48 | 15.13 | 35.62 | 21.1 | 41.99 |
| Germany | | | | | | | | | | |
| | Avg pos | Avg neg | No sent | Strong sent | Strong pos sent | Strong neg sent | Pos sent only | Neg sent only | Pos > neg | Neg > pos |
| **Base** | 0.385 | -0.604 | 46.91 | 6.01 | 0.5 | 5.58 | 18.84 | 23.31 | 20.14 | 28.34 |
| **Phase 1** | 0.347*** | -0.637** | 48.80 | 5.50 | 0.34 | 5.18 | 16.68 | 24.14 | 17.66 | 29.7 |
| **Phase 2** | 0.375***˚˚ | -0.506***˚˚˚ | 51.13 | 3.84 | 0.47 | 3.39 | 19.56 | 20.35 | 20.68 | 24.42 |
| **Phase 3** | 0.343***˚˚˚ | -0.542***˚˚˚ | 50.79 | 3.9 | 0.4 | 3.51 | 17.65 | 22.57 | 18.74 | 26.79 |
| The Netherlands | | | | | | | | | | |
| | Avg pos | Avg neg | No sent | Strong sent | Strong pos sent | Strong neg sent | Pos sent only | Neg sent only | Pos > neg | Neg > pos |
| **Base** | 0.424 | -0.820 | 35.95 | 9.11 | 1.19 | 8.05 | 13.07 | 34.31 | 16.71 | 39.82 |
| **Phase 1** | 0.450* | -0.879*** | 31.37 | 7.4 | 0.99 | 6.59 | 13.28 | 37.54 | 17.22 | 43.31 |
| **Phase 2** | 0.524***˚˚˚ | -0.874*** | 29.84 | 8.3 | 1.54 | 6.85 | 15.76 | 34.07 | 20.22 | 40.79 |
| **Phase 3** | 0.535*** | -1.022***˚˚˚ | 25.64 | 10.89 | 1.87 | 9.21 | 13.88 | 37.96 | 18.51 | 45.99 |
| Sweden | | | | | | | | | | |
| | Avg pos | Avg neg | No sent | Strong sent | Strong pos sent | Strong neg sent | Pos sent only | Neg sent only | Pos > neg | Neg > pos |
| **Base** | 0.505 | -0.933 | 22.30 | 5.53 | 0.92 | 4.69 | 11.92 | 37.41 | 16.12 | 45.62 |
| **Phase 1** | 0.515 | -1.003* | 14.16 | 5.97 | 1.19 | 4.95 | 11.09 | 44.88 | 15.52 | 50.17 |
| **Phase 2** | 0.552*** | -0.964 | 18.17 | 5.05 | 1.09 | 4.04 | 12.3 | 38.44 | 17.79 | 46.35 |
| **Phase 3** | 0.599*** | -1.047***˚˚ | 13.93 | 6.98 | 2.48 | 4.59 | 10.67 | 40.13 | 16.4 | 49.3 |

Two-sample t-test compared to base set

* = p<0.05 compared to base set

** = p<0.01 compared to base set

*** = p<0.001 compared to base set

˚ = p<0.05 compared to the previous phase

˚˚ = p<0.01 compared to the previous phase

˚˚˚ = p<0.001 compared to the previous phase

Interpretation

No sentiment: pos = 0 AND neg = 0

Strong sentiment: pos = [3;4] OR neg = [3;4]

Strong positive sentiment: pos = [3;4]

Strong negative sentiment: neg = [3;4]

Only positive sentiment: pos = [1;4] AND neg = 0

Only negative sentiment: pos = 0 AND neg = [1;4]

Positive score larger than neg: pos>neg

Negative score larger than pos: neg>pos

topic modeling results based on short texts such as tweets tend to include considerable noise (i.e. words not informative about the identified topic) [104–106], we cross-referenced our findings by manually looking up the tweets in question to ensure that our interpretations of the topics were accurate. As a complement to the topic modeling analysis, we consider corpus statistics (prominent words) and bigrams where appropriate (see bigram and corpus statistics results in *S4 Appendix: bigrams and corpus statistics*).

## Results

All sentiment analysis results may be found in Table 4 below. For the raw sentiment analysis data, please refer to *S5 Appendix*: *Sentiment analysis results*. All topic modeling analysis tables may be found in *S6 Appendix*: *Topic modeling results*. In the following section, the results of the sentiment analysis and topic modeling analysis will be presented by country followed by a comparative analysis.

### Denmark

Denmark initially experienced a negative turn in sentiment with APS dropping from 0.613 (base levels) to 0.503 ($p \leq 0.05$ (two-samples t-test)), and ANS rising from -0.805 to -0.997 ($p < 0.001$). As the country entered phase 2, however, APS rose to 0.650, while ANS differed significantly from neither base levels nor phase 1. In the third phase, sentiment seemed to have stabilized: APS fell to 0.618, which is statistically significantly different to neither base levels nor the previous phase, while ANS stayed around phase 2 levels, which is slightly higher than base levels.

Corpus statistics and bigrams revealed the presence of considerable levels of uncertainty and anxiety. The topic modeling analysis showed that people worried about the severity of Covid-19, the country's intensive care capacity, and how to protect vulnerable citizens, (Phase 1, topics 4, 10, and 13). In addition, the government and the prime minister were accused by both experts and laymen of inciting panic (7) and attempting to score cheap political points by appearing decisive (15).

As reflected in the sentiment analysis, however, public opinion on Twitter shifted in a positive direction with Phase 2: the second and third most prominent topics consisted mainly of messages thanking or praising the prime minister Mette Frederiksen and the government (the most prominent topic consisting mainly of noise (i.e. words without topical meaning). Compared to Phase 1, topics relating to uncertainties also diminished; instead, political negotiations (phase 2, topics 5, 10, and 15) and political messages (6, 7, 8, 9, and 14) were prominent topics, especially those relating to containment measures (6, 7, 8, 13, 15). Criticism in this phase was similar to that of the previous phase, although slightly more severe: the government was described, not just as poor communicators (17), but as deceptive (19), and not just as being opportunistic, but as poor leaders (17) and, in relation to a piece of emergency legislation allowing the police to enforce compulsory treatment for Covid-19 victims, as violating democratic rights (15). This is peculiar, as not only did ANS not increase during phase 2, but both the number of tweets with a higher negative than positive sentiment score and the number of tweets with strong negative sentiment fell. This, then, indicates that the number of dissatisfied Twitter users diminished, but that those that remained, were either more pointed in their criticism or, conversely, more ambiguous (i.e. having sentiment scores with equally strong positive and negative sentiment). While some Twitter users thus because more radicalized in their view as the crisis continued, the results indicate that many also became more nuanced.

While the third phase saw no statistically significant changes in sentiment score, the topic modeling analysis included fewer words of praise toward Frederiksen or the government; most praises were given in reaction to critical comments (phase 3, topic 14). Rather, discussion was centered around reopening efforts (1, 3, 5, 6, 7, 11, 20) with a focus on children returning to school (5, 20) and vulnerable citizens (3, 7). Criticism did not differ thematically from the previous phase: the government was reproached for their strategy, which was perceived to be unilateral and excessive (13), as well as for political opportunism (10).

## Germany

Germany experienced a negative turn in sentiment in Phase 1: the APS of the phase dropped from 0.385 (base levels) to 0.347 (p<0.001), and the ANS rose from -0.604 to -0.637 (p≤0.01). This pivoted into a positive turn in the second phase, however: APS rose to 0.375 (p≤0.01) ANS dropped to -0.506 (p<0.001). In Phase 3, the sentiment scores reverted to phase 1 levels, although with a lower ANS: APS dropping to 0.343 (p<0.001 compared to phase 2), and ANS increased slightly to -0.542 (p<0.001 compared to Phase 2).

Fear and uncertainty characterized Phase 1: the corpus statistics included prominent words like *spread*, *measures*, *outbreak*, *panic*, *anxiety*, *economy*, and *worries*. As Germany had its first case of Covid-19 as early as end-January, the initial patterns of contagion were still in the public eye (Phase 1, topics 4, 5, 6, 8, 9, 13, 15, 17). Twitter users did not express much faith in the German Chancellor Angela Merkel during this phase and called for her and the government to be more proactive and communicative (11, 14, 16). Some characterized Merkel as incompetent (10), and conspiracies accusing the chancellor of being in cahoots with China as the covert creators of Covid-19 even gained some traction (18, 19).

In Phase 2, fear and uncertainty seemed to diminish: while prominent bigrams and the corpus statistics still contained indicators of uncertainty (*measures*, *crisis*, *battle*, *economy*), this phase was characterized to a higher degree by expressions of solidarity and social awareness (*curfew*, *agreement*, *#Stayathome*, *solidarity*, *positive*, *#wirbleibenzuhause*). The focus of German Twitter had moved toward political communication (Phase 2, topics 3, 8, 11, 14, 17)–especially statements by Merkel, e.g. the statement "Es ist ernst" ("This is serious") and the description of the Covid-19 crisis as the biggest challenge since World War II–as well as concrete political decisions (5, 6, 8, 9, 19, 20). While praise of Chancellor Merkel or the government did not make up a separate topic, words like solidarity or the hashtags #StayAtHome or #wirbleibenzuhause (#westayathome) were widely used in conjunction with the retweeting of Merkel's statements, as was the bigram "Danke Merkel". Criticism mainly related to a continued desire for the government to be more proactive (2, 15) and more communicative (16).

In Phase 3, focus continued to be on political communication, in particular Merkel's so-called "Öffnungsdiskussionsorgien" (opening discussion orgies)(Phase 3, topics 4, 13), with which Merkel had characterized the discourse in the individual federal Länder, and which many Twitter users felt was inappropriate in tone (13). The biggest topic of discussion concerned the reopening of the country post-lockdown (3, 4, 5, 7, 9, 10, 13, 16, 17, 20), which was either seen as necessary, and the government as having gone too far already (14, 16, 20), or as premature (3, 4, 13, 17). The portrayal of Merkel and the government was mostly critical (topics 16, 20), and with phrases like "Danke Merkel" and "Danke Regierung" being almost absent. In addition, the conspiracy topic from Phase 1 resurged (topic 12). Finally, the tone of solidarity found in Phase 2 greatly diminished in this phase: *Solidarity* went from being used 346 times across 26107 tweets (1.3% of tweets, assuming one mention per tweet) to 55 times across 7270 tweets (0.7% of tweets, assuming one mention per tweet), #wirbleibenzuhause from being used in 1% of tweets to 0.4%, and #StayHome from 0.4% to 0.3%.

## The Netherlands

Dutch Twitter was characterized by increasing polarization in sentiment as the crisis progressed. The proportion of no-sentiment tweets fell consistently over time from 35.95% to 25.64%, and APS and ANS both became progressively stronger. The first phase was characterized by a small, but significant increase in polarization: APS rose from 0.424 (base levels) to 0.450 (p≤0.05), and ANS rose from -0.820 to -0.879 (p<0.001). Phase 2 subsequently saw a

rise in APS to 0.524 (p>0.001), ANS not changing significantly, whereas Phase 3 saw a strong increase in ANS from -0.874 to -1.022, while APS stayed at the same level.

The bigrams and corpus statistics of this phase indicated that the Dutch Twitter population was on high alert (necessary, quarantine, infections, deaths, crisis), which was further corroborated by the topic modeling analysis (Phase 1, topics 1, 3, 5, 6, 10, 14, 15, 17). The phase was characterized by a strong call for political action (1, 13, 15): in particular, the lack of immediate introduction of containment measures had Twitter questioning the priorities and attitude of the Minister-President Mark Rutte and the government (7, 12, 15, 17), and some worry regarding the vulnerable population was expressed (2, 6). Two critical articles, "Rutte has the wrong priorities" (Telegraaf.nl) and ""82 corona patients in the Netherlands: ultra-screw-up Rutte has completely lost control" (Dagelijkse Standaard) were in heavy rotation (see also [107, 108]).

The major conversation theme of Phase 2 was Rutte's address to the nation on March 16[th] (Phase 2, topics 2, 4, 10, 11, 17). People particularly criticized Rutte's remarks on the effectiveness of herd immunity (3, 19), and many perceived the decision as Rutte prioritizing the economy above human lives (5). Conversely, the word *good* was used commonly in relation to the March 16[th]; as such, the rise in APS may be due to many users being satisfied with Rutte's speech. In addition to the March 16[th] address, outbreak containment measures such as social distancing, the closing of schools, and whether the country should close borders, were discussed (7, 12, 13) with people expressing worry about infectivity and intensive care resources (8, 16).

In Phase 3, containment measures–especially face masks and corona tracking apps (Phase 3, topics 2, 4, 7, 13, 18, 19)–continued to be widely discussed (2, 4, 10, 18), as was the continued threat of Covid-19 (1, 5, 8, 17, 19). Rutte was the subject of some criticism, although this was both less prominent than in the previous phase and included people believing the eventual lockdown to have been too excessive (9, 20).

## Sweden

Phase 1 in Sweden saw a slight, but significant increase in ANS compared to base levels from -0.933 to -1.003 (p≤0.05), with no significant difference in APS. Phase 2 saw no statistically significant change in sentiment compared to Phase 1, which may be explained by the fact that no new policy was introduced until far into the second phase; nevertheless, there was a significant increase in APS compared to base levels. In phase 3, however, ANS dropped from -0.964 to -1.047 (p≤0.01) with no significant change in APS.

Most topics during Phase 1 were related to political performance (Phase 1, topics 2, 4, 5, 6, 7, 8, 9, 10, 14, 18) and political communication, especially press briefings (4, 7, 9). Public opinion on Twitter was generally unfavorable, with the government being described as incompetent (5), opportunistic (18) or gambling with people's health (15, 16). Call for political action was also prevalent with many Twitter users questioning keeping the borders open (3, 10) and allowing travel to high-risk zones.

During the second phase, the word *good* was very prominent (165 mentions across 2595 tweets), which, together with the rise in APS, indicates a positive shift in public opinion on Twitter. The major themes of the phase were new policies–a business bailout (Phase 2, topic 4), remote classes for high schools and universities (7), and a ban on gatherings of more than 500 people (6)–and criticism of the government centered around them not doing enough (9, 12, 15, 17), especially for the vulnerable population (9, 12). Calls for political action became more concrete, with users calling for a country-wide quarantine (8), better healthcare equipment (9), and a ban on refugees from high-risk zones.

The topic modeling analysis suggests that this negative turn in sentiment in the third phase was motivated by a high mortality rate: not only was the Swedish death toll high (Phase 3, topics 13, 18), but it was notably higher than those of neighboring countries (10). The topic modeling analysis further indicates a waning trust in the government: calls for political action and political transparency continued (4, 6), the government was described as failed or having the wrong priorities (11, 19), and a statement by the retired virologist and researcher Jan Stillström, calling the main designer of the Swedish Covid-19 strategy, the state epidemiologist Anders Tegnell, a "politically appointed fraud", was circulated.

## Discussion

As may be expected when comparing four highly similar countries with very different crisis responses, the analysis showed both similarities and variation in the population response. Overall, the development in sentiment scores among the four countries may be characterized as similar in outset with increasingly divergent paths. All countries experienced an initial negative turn accompanied by uncertainty and anxiety as well as an appeal to the government to act on the crisis, followed by a positive turn in the weeks immediately after the classification of Covid-19 as a pandemic by the WHO, which corpus statistics and bigrams indicated to be a function of the people rallying around their governments. At this point, Denmark and Germany followed similar paths, having particularly strong positive turns that evened out in Phase 3. By comparison, Dutch sentiment was alone in experiencing significant increases in both APS and ANS during Phase 1, increasing even more in polarization in the subsequent phases, whereas Sweden stayed comparatively stable until Phase 3, in which Swedish ANS increased markedly, culminating in a polarization of sentiment similar to that of the Dutch sentiment results.

Looking at the similarities in sentiment scores among the four countries, i.e. the initial negative turn, the positive turn in Phase 2, and the final move toward more neutral sentiment scores in Phase 3, it seems to be the case that a large part of public sentiment is a direct result of the impending pandemic and not majorly affected by any political decisions: the initial uncertainty of the situation created anxiety in the population as well as a call for action, which turned into support for the authorities (the so-called "rally around the flag" effect [109]), as containment plans were being rolled out, and after people started to become fatigued support waned (cf. [32]). Concretely, this means that, on the one hand, no matter how skilled authorities may operate, they may not be able to eliminate uncertainty and anxiety in the population when initially faced with a pandemic; on the other hand, no matter the response, authorities in high-trust countries are likely to find support in the population during the initial steps of a containment strategy.

Still, the variations in sentiment we found between the countries seem to be largely in line with Boin et al.'s [30] observations on authority credibility during a crisis: the two countries with the most assertive responses–Denmark and Germany–experienced the strongest positive turn in Phase 2, indicative of stronger support in the population. And, while a rise in APS in Phase 2 of the Swedish results indicates that the laissez-faire strategy of the Swedish authorities initially garnered some support, the subsequent rise in ANS in Phase 3 points toward a rising dissatisfaction, which could be explained by the drawbacks of the Swedish strategy becoming more visible (cf. [110–112]). An interesting nuance to the observations of Boin et al. [30] is that both the Dutch and the Swedish sentiment results, rather than simply having more negative and less positive valence (compared to Denmark and Germany), were, in fact, stronger on both valences–and thus indicative of a polarization of opinion taking place. A possible interpretation of this that when authorities respond to a crisis by implementing strict

countermeasures, the population–regardless of individual thoughts on the matter–tempers its emotional reaction and puts at least some amount of trust in the official strategy (cf Maor's "Regulatory overreaction" [113, 114] and Van Wijk and Fischhendler's "urgency discourse" [115], whereas a slower or more ambiguous stance on the part of the government may fail to temper emotional reactions to the crisis [116–118]. A possible explanation is that a prompt reaction by the government to a major crisis addresses the shock induced by the social disruption element of the crisis rather than the expected consequences of it, as, for example, a crisis-induced recession may cause damages equal in scope. As the countries in this study are not sufficient for a large scale generalization, however, the effect of crisis response on population sentiment is a question for future studies.

The topic modeling analysis informs the findings of the sentiment analysis. Throughout Phase 1, criticism of authorities was prominent in all countries; nevertheless, the type criticism directed at the authorities differed depending on the country. While the Danish and German governments were criticized for a lack of communication, the Swedish and Dutch governments were criticized for treating citizens' lives with recklessness. Swedish leaders were further accused of being both incompetent–an accusation they shared with the German leaders–and politically opportunistic–an accusation, which the Danish leaders were similarly subject to. Authorities managing a crisis may thus benefit from the knowledge that the amount of criticism they are subject to during the initial stages of a crisis response may be less an expression of the reception of their strategy in the population and more an inevitable expression of uncertainty.

During the second phase the authorities of all countries experienced both positive and negative feedback, with Danish and German Twitter as the most supportive: two of the most prominent Danish topics were identifiable as praise toward the government and Mette Frederiksen, and Germany had the bigram "Danke Merkel" showing up a total of 132 times across 26107 tweets, and words like solidarity, positive, and #Stayathome were used in conjunction with Merkel's statements. The possibility mentioned earlier in relation to the Swedish Phase 2 sentiment upswing that the image of authorities as in control is initially more important than concrete measures was further supported, as focus in the German tweets was more on political communication and a sense of community than on the containment measures themselves.

Criticism directed at authorities mostly continued along the path set in Phase 1, focusing on passivity and lack of communication. Criticism of the Swedish authorities leaned more to the side of incompetence, challenging the effectiveness of "herd immunity". Looking at the differences in upswing between Germany (slight increase in APS, strong decrease in ANS) and the Netherlands (moderate increase in APS, no changes to ANS), the importance of message timing, as according to Boin et al. [30], was found to be accurate. Although the lockdown in The Netherlands started at the same time as in Germany, communication had until then hinted at following a similar containment strategy as in Sweden. Subsequently, tweets in the Netherlands appear to place less emphasis on solidarity and cooperation in their references to adherence to government containment measures than in Germany. The results seem to indicate that early political communicative action is key to establishing a sense of solidarity and personal responsibility for the successful handling of the crisis among citizens [116].

In Phase 3 we observe a decrease in overt support for the authorities: only one topic was dedicated to praise or thank you messages (DK: topic 14). The most prominent topics in Denmark and Germany concerned for example reopening efforts. This "neutral turn" might be explained by frustrations surrounding lack of communication from the authorities–a common complaint in all countries. Indeed, in accordance with existing theory on crisis communication (esp. (esp. [33]) some people turned to alternative information sources and explanations. We see topics relating to conspiracy theories appearing as well as articles featuring experts

criticizing the government. An interesting aspect not discussed in existing theory was observed, however. The two countries with assertive authority reactions to the crisis, i.e. Denmark and Germany, were also the two countries, in which complaints of poor communication from the authorities showed up in the topic modeling analysis. This implies that transparency is important, not only regarding risk but also regarding strategy. Even if authorities are still considering the best approach, this may thus be beneficial to communicate explicitly to the population.

## Conclusion

This study analyzed the public perceptions of governmental authorities in Denmark, Germany, the Netherlands, and Sweden during the first three phases of the COVID-19 pandemic as found on Twitter. Supplementing earlier studies studying tweets in English [19, 20], it indicated that populations in high-trust countries respond more favorably to swift and extensive crisis strategies by visible and communicative authorities. In line with existing theory on the topic, the Twitter populations of the four countries under investigation expected their authorities to show capacity to act, to be transparent in their communication with regards both to the risk of the virus and the considerations behind the countermeasures taken, and to not make light of the situation.

The major points of existing disaster research on the effects of authority response to crisis situations on population sentiment and discourse were overall shown to be applicable to high-trust countries as well. As suggested by Boin et al. [30] and Schneider [32], the respective populations did expect authorities to take an active role in crisis management: a more assertive crisis response was generally well-received in the population, whereas attempts to maintain a more normal everyday life for the citizens caused at least some amount of anxiety. Furthermore, signaling a capacity to act early in the process seemed to create a more favorable response, even if the measures implemented were the same. Finally, in accordance with existing research on information seeking during disasters, articles critical of the government line were widely circulated in all countries, and conspiracy theories were commonplace [32, 37, 38].

Nevertheless, this study, while needing to be further verified by dedicated studies, also provided some interesting nuance. The countries with the least assertive strategies, rather than a clear-cut negative response, experienced a heightened degree of polarization of sentiment. An assertive response then, rather than merely lending credibility to authorities, seems to prevent division in the population. Even with the resulting polarization, however, the results indicated a wide and prolonged "rally around the flag" effect [109]–a notable result in the case of Sweden, which stood alone on the global stage by not having a lockdown comparable to those of other countries–which may be attributable to the high trust of the analyzed countries [119].

The findings of this study are relevant to policy makers and, especially, visible authorities in high-trust countries during epidemic events. Corroborating existing theory, this study demonstrates the importance of clear, comprehensive, and timely communication, not only with regards to the epidemic at hand, but with regards to deliberations surrounding strategy as well. In addition, policy makers utilizing a less proactive approach may benefit from anticipating political polarization in the population [120, 121].

Of course, this study is not without its limitations. First, there is currently little research on the demographic makeup of twitter and other social media in European countries, and thus overall representativeness is difficult to determine. Second, both sentiment analysis and topic modeling analysis are still in their infancy and, having been initially developed for the English languages, still are at their best when processing English texts; NLP software developers should thus improve on the capabilities of these programs to improve non-English language

processing. Finally, it is important to keep in mind that this study was performed with a limited number of countries. Future research should confirm the findings of this study using alternative methods as well as replicate the study with other countries, especially countries with different levels of prior governmental trust or with a different severity of the COVID-19 outbreak than the countries of this study. In addition, the limited time frame of the study meant that the notion of fatigue from outbreak containment measures was only briefly touched upon. Future research should thus continue to investigate population response to governmental containment strategies, in the long term and during the second wave.

## Supporting information

**S1 Appendix. Timelines (by country).** Includes Tables A through D.
(PDF)

**S2 Appendix. Twitter search terms.** Includes a single table, Table A.
(PDF)

**S3 Appendix. Sentiment dictionaries (DK and SE).** .zip-file including the Danish and Swedish SentiStrength dictionaries developed for the purposes of this study.
(ZIP)

**S4 Appendix. Bigrams and corpus statistics.** Contains two.exe-files: part 1 of 2 (DK, DE) and part 2 of 2 (NL, SE).
(ZIP)

**S5 Appendix. Sentiment analysis results.** Includes results of the sentiment analysis including sample sizes before and after cleaning, average positive score (APS) with standard deviance, average negative score (ANS) with standard deviance, and sentiment tables (Figs) A-0 through -D3, which portray the distributions of tweets by sentiment score per country per phase.
(PDF)

**S6 Appendix. Topic modeling results.** Includes Tables A1-D3 with the results of the topic modeling analysis.
(PDF)

**S1 File.**
(DOCX)

## Acknowledgments

The authors would like to thank Dr. Koen van Eijck, who acted as a second reader, when this article was still in the form of a master's thesis. We would also like to thank everyone who are showing consideration toward others by practicing social distancing, wearing face masks and working from home during the COVID-19 pandemic.

## Author Contributions

**Conceptualization:** Sara Grøn Perlstein, Marc Verboord.

**Data curation:** Sara Grøn Perlstein.

**Formal analysis:** Sara Grøn Perlstein.

**Investigation:** Sara Grøn Perlstein.

**Methodology:** Sara Grøn Perlstein, Marc Verboord.

**Resources:** Sara Grøn Perlstein.

**Supervision:** Marc Verboord.

**Validation:** Sara Grøn Perlstein.

**Visualization:** Sara Grøn Perlstein.

**Writing – original draft:** Sara Grøn Perlstein.

**Writing – review & editing:** Sara Grøn Perlstein, Marc Verboord.

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
