## [Decision Letter · Decision Letter 0]

25 Feb 2021

PONE-D-21-02692

Lockdowns, lethality, and laissez-faire politics. Public discourses on authority in high-trust countries during the COVID-19 pandemic.

PLOS ONE

Dear Dr. Perlstein,

Thank you for submitting your manuscript to PLOS ONE. After careful consideration, we feel that it has merit but does not fully meet PLOS ONE’s publication criteria as it currently stands. Therefore, we invite you to submit a revised version of the manuscript that addresses the points raised during the review process.

We look forward to receiving your revised manuscript.

Kind regards,

Prof. Anat Gesser-Edelsburg

Academic Editor

PLOS ONE

Journal Requirements:

2. PLOS ONE has specific requirements for studies using personal data from third-party sources, including social media, blogs, other internet sources, and phone companies (https://journals.plos.org/plosone/s/submission-guidelines#loc-personal-data-from-third-party-sources). These requirements include confirming data are collected and used in accordance with the company or website’s Terms and Conditions, obtaining appropriate ethics or data protection body review, and ensuring appropriate consent from individuals whose data are used in research. In this case, please ensure that your Ethics statement is in compliance with guidelines, and that you have complied with the company's (i.e., Facebook's) Terms and Conditions, with appropriate permissions.

Reviewers' comments:

Reviewer's Responses to Questions

**Comments to the Author**

1. Is the manuscript technically sound, and do the data support the conclusions?

Reviewer #1: Partly

Reviewer #2: Partly

2. Has the statistical analysis been performed appropriately and rigorously? 

Reviewer #1: I Don't Know

Reviewer #2: I Don't Know

3. Have the authors made all data underlying the findings in their manuscript fully available?

Reviewer #1: Yes

Reviewer #2: No

4. Is the manuscript presented in an intelligible fashion and written in standard English?

Reviewer #1: Yes

Reviewer #2: Yes

5. Review Comments to the Author

Reviewer #1: Commendable effort in conducting the study and putting this manuscript together. Perhaps, it could be better if the section on 'Data and Method' (i.e. line 224) and the section on 'Data' (i.e. line 232) could be combined as 'Data Collection' and placed as sub-sections under the section on 'Methodology' (i.e. line 266). Moreover, the section title 'Results and Discussion' should be changed to 'Results' as there is already a section on 'Discussion' later on. Please also revisit the References and try to be consistent about the style of referencing. It is an interesting study. Well tried.

Reviewer #2: Comments:

General - Needs a review for grammar/ punctuation.

Introduction - The first paragraph merits many more citations - currently, there are none.

Introduction - I think there is an important difference between trust in government and trust in authorities - the two ideas are discussed interchangeably here, but I think they merit teasing out if possible. For example, in the US there is a difference between mistrust in the CDC/ FDA and mistrust in the Trump/ Biden administrations. They’re connected, but also separate. Someone can have low trust in a political administration but high trust in a health authority.

Line 36 - What are the specific dates of these phases? Would be helpful to understand your analysis periods. For example, what does “some time” mean?

Line 45 - dangling parenthesis

Line 61 - A citation for this statement is needed

Line 65 - Personally, I don’t see the need for an entire “literature review” section, unless requested by the journal. I think much of this woudl be better placed in an introduction, before you state the aims of the paper because it sets the paper up to flow nicer. After I read the introduction, I had a few questions that appear to be answered in the literature review section. I also think the literature review section can be condensed quite a bit, with some of the sentences that aren’t specifically relevant (like the part on ebola in West/Central Africa) just included as a citation, and not given so much prominence. This will make it easier to merge the literature review section into the introduction.

Line 91 - Skeptical

Line 114 - This seems too drastic, and the sentence is a bit confusing following the previous one (which says the population is more forgiving of an exaggerated response): “but is bound to ultimately result in failed expectations.” This sentence seems out of order in the paragraph

Line 160 - Are there other citations that you can use to support this paragraph? This and the one before only have 2 citations, but I imagine that there is a larger body of evidence that you can cite here, particularly if they are more relevant to the topic of the paper.

Line 172 - Similar to above, I think you can condense this section quite a bit - especially given that you include more details in the appendix.

Line 177 - I dont think that you’ve established how you determined that there is a high level of trust in each of these countries. Is this based on a citation? A hunch? I’d like to see more information about how you determined trust in each of the 4 countries, given the topic of the paper.

Line 225 - Dates would be helpful in this sentence, to understand what your points in time were. Also, is there a justification for the separation into these points, or is it just based on your own intuition? There are also many appendices here - you can remove Appendix C in particular if you just put the dates in text.

Line 232 - Do you have a citation for GetOldTweets? What kind of access to Twitter does this service have? 100% of publicly available tweets? Did the service provide you with metadata to geotag posts? Have other researchers used this service before? More information about the specific software platform would be useful.

Line 253 - It looks like you searched for political terms - so my comment above about “health authorities” is not relevant - I’d just make that clear at some point here. When you are referring to “authorities” you are referring specifically to political authorities, not health authorities.

Line 262 - How did you check to ensure that this method worked to geotag posts? Did you manually code a subset of posts? To say that it “should ensure” is not a strong enough justification for the methods.

Line 263 - Do you have any citations of others who have adopted this approach and shown it to have merit, given its limitations? Likewise, did you rely on other research to help you create the search query, or to guide your analysis more broadly?

Line 264 - Similar to above, Appendix E can be worked into the text and removed as an appendix.

Line 276 - For the example here, instead of marshmallows can you provide an example that is drawn from the research?

Line 283 - I don’t see an Appendix F in the materials submitted.

Line 284 - Were all tweets required to have a sentiment? Were there instances in which you were unable to determine a sentiment and they were excluded from analysis? Were any tweets excluded from analysis?

Line 291 - There is only one citation for the topic modeling analysis, but it would be good to see more - have others used this strategy before? What other work did you consult in creating this approach?

I think that Tables 2 A - L are better suited for an appendix.

Line 483 - I wouldn’t say that “it is clear” - In general, I think the conclusions seem to express too much certainty. I think your results suggest many of these statements, but particularly given that Twitter is not representative of the larger population, I think many of the statements need to be toned down.

Line 502 - Capitalization / grammar error

Line 504 - Do you have any citations - even news stories - that support these ideas? Either from the pandemic, or from previous examples? Similar question for the following paragraph. I’d like to see more citations support the findings, to place the findings within information that has already been published.

Line 513 - “Playing fast and loose” seems too colloquial

Line 517 - “May be less an expression” vs “is less an expression”

Line 567 - Citation for this

Line 584 - I’m not sure that you have sufficient evidence to say that conspiracy theories were not as impactful as existing literature - first I think you need to cite the literature, and then I think you need to dig into this more. Why do you say they aren’t as impactful? The following sentence does not provide any support for this statement.

6. PLOS authors have the option to publish the peer review history of their article (what does this mean?). If published, this will include your full peer review and any attached files.

Reviewer #1: No

Reviewer #2: No

---

## [Author Response · Author response to Decision Letter 0]

14 Apr 2021

Response to Reviewers

First of all we want to thank the editor for the opportunity to revise the manuscript. We would like to thank the reviewers and the editor for their insightful and constructive feedback. This has helped us to improve the paper considerably. 

Please find the responses to the points raised by the editor and reviewers below.

Editor

1.Style requirements

RESPONSE: We have made sure the manuscript follows the style requirements.

2.Ethics

RESPONSE: We have added an ethical statement in the methodology section. 

3.Include captions

RESPONSE: We have included captions for our supporting information.

Reviewer #1

Commendable effort in conducting the study and putting this manuscript together. Perhaps, it could be better if the section on 'Data and Method' (i.e. line 224) and the section on 'Data' (i.e. line 232) could be combined as 'Data Collection' and placed as sub-sections under the section on 'Methodology' (i.e. line 266). Moreover, the section title 'Results and Discussion' should be changed to 'Results' as there is already a section on 'Discussion' later on. Please also revisit the References and try to be consistent about the style of referencing. It is an interesting study. Well tried.

RESPONSE: We want to thank the reviewer for the compliments and the useful feedback.

We agree with the suggested changes in the structure and have adjusted the manuscript accordingly. 

Structure in the Data and Methods section changed to:

Methodology

Data collection

Method of analysis

Results and Discussion section changed to “Results” and the “Discussion” subsection changed to a main section.

Reviewer #2

General - Needs a review for grammar/ punctuation.

RESPONSE: We have reviewed the manuscript on this and improved grammar/punctuation.

Introduction - The first paragraph merits many more citations - currently, there are none.

RESPONSE: We agree with the reviewer that more references are warranted. We have added multiple references, both with regards to COVID-19 and to the issue of trust.

Introduction - I think there is an important difference between trust in government and trust in authorities - the two ideas are discussed interchangeably here, but I think they merit teasing out if possible. For example, in the US there is a difference between mistrust in the CDC/ FDA and mistrust in the Trump/ Biden administrations. They’re connected, but also separate. Someone can have low trust in a political administration but high trust in a health authority.

RESPONSE: We agree with this comment. Our paper focuses on trust in governments or political trust, not trust in health authorities. We have clarified this. We have added “political” when we mention authorities throughout the article to ensure there is no misunderstanding.

RESPONSES TO SPECIFIC COMMENTS PER LINE:

Line 36: RESPONSE: changed “some time” to “a month”. Dates are added.

Line 45: RESPONSE: we corrected dangling parenthesis.

Line 61: RESPONSE: we added citations.

Line 65: Personally, I don’t see the need for an entire “literature review” section, unless requested by the journal. I think much of this woudl be better placed in an introduction, before you state the aims of the paper because it sets the paper up to flow nicer. After I read the introduction, I had a few questions that appear to be answered in the literature review section. I also think the literature review section can be condensed quite a bit, with some of the sentences that aren’t specifically relevant (like the part on ebola in West/Central Africa) just included as a citation, and not given so much prominence. This will make it easier to merge the literature review section into the introduction. 

RESPONSE: We see the point of the reviewer that the current literature review is strictly speaking not a literature review and might therefore be confusing. At the same time, we think it is relevant to provide theoretical background and it appeared difficult to fully include this in the introduction. We have followed the example of Wicke & Bolognesi (2020), which is cited, and changed the section name to “Theoretical background”. We hope this is acceptable for the reviewer. Furthermore, we followed the advice of the reviewer to take out certain parts (incl. parts on ebola) which were not specifically relevant. 

Line 91: RESPONSE: we changed ‘skeptic’ to ‘skeptical’.

Line 114: This seems too drastic, and the sentence is a bit confusing following the previous one

RESPONSE: we changed “is bound to” to “may”. If we understand the reviewer correctly, they felt like the statement that “the population may even be more forgiving of an exaggerated response than an insufficient one” was contradicted by the statement that “too much confidence may actually be harmful in the long run”. Since response refers to actions and confidence to attitude (the word rhetoric is mentioned in the subsequent section as well), the intent was to signal that these statements refer to different aspects of the crisis response. In order to highlight the difference, we changed the word “response” in the former statement to “strategy”. If we misunderstood the comment, we are, of course, open to change this. 

Line 160: RESPONSE: we added more references.

Line 172: RESPONSE: We have shortened this section a bit, although we do want to highlight a couple of substantial characteristics of the contexts in various countries. 

Line 177: I dont think that you’ve established how you determined that there is a high level of trust in each of these countries. Is this based on a citation? A hunch? I’d like to see more information about how you determined trust in each of the 4 countries, given the topic of the paper. 

RESPONSE: We base ourselves on background literature. The reviewer is correct that more references are needed to support this statement. In the new version the description of trust has changed: 

“Statistical data consistently shows these four countries to have a high level of population trust in their authorities (OECD, 2020; Eurobarometer, 2015; Mungiu-Pippi et al., 2015; Our World in Data, 2018; The World Bank, 2020); as such, these four countries provide a valuable case study of the effects of different containment strategies on public opinion in high-trust countries.”

Line 225: RESPONSE: We added dates by bringing Appendix C into the main text (table 1). We provide an argumentation for these dates.

Line 232: RESPONSE: We have added a more detailed description of GetOldTweets, incl. sources on the software as well other researchers using it.

Line 253: RESPONSE: Thank you for the observation. Indeed, it may not have been clear that we were referring specifically to political authorities throughout the paper. We have corrected this throughout, and it should be clear now. 

Line 262: REPONSE: We have added more details on geographical allocation.

Line 263: RESPONSE: We think the additional details should cover this point but let us know in case more information is needed. 

Line 276: RESPONSE: “Marshmellows are healthy” is changed to “Merkel is a competent leader”. 

Line 283: RESPONSE: Apologies, the appendices were not correctly named. We adjusted this.

Line 284: RESPONSE: Details on no-sentiment tweets have been added. In short: they are included and simply have a value of 0 when calculating the avg sentiment. 

Line 291: RESPONSE: Some more info on topic modelling on short texts have been added. Some extra references and a little more detail at the beginning of the “Method of analysis” section have been added. 

Line 483: RESPONSE: We agree that the conclusions at times express too much certainty. We have rewritten the conclusion. “it is clear” has been changed to “it seems to be the case”. 

Line 502: RESPONSE: We have corrected this; ‘of’ added after regardless

Line 504: RESPONSE: We have added sources.

Line 513: RESPONSE: We agree and have changed “playing fast and loose with citizens’ lives” to “treating citizens’ lives with recklessness”. 

Line 517: RESPONSE: We changed ‘may be less an expression’ to ‘is less an expression’

Line 567: RESPONSE: We added a citation.

Line 584: RESPONSE: We agree and have the conclusion on conspiracy theories removed as not really central to the main thesis of our article. 

Appendices moved around according to the notes of R2. 

Appendix B -> A 

Appendix D -> B

Appendix A -> C

Appendix H -> D

Appendix G -> E

Appendix with TM results -> F (indeed, this was missing from the uploaded data – our apologies)

---

## [Decision Letter · Decision Letter 1]

11 May 2021

PONE-D-21-02692R1

Lockdowns, lethality, and laissez-faire politics. Public discourses on political authorities in high-trust countries during the COVID-19 pandemic.

PLOS ONE

Dear Dr. Perlstein,

Thank you for submitting your manuscript to PLOS ONE. After careful consideration, we feel that it has merit but does not fully meet PLOS ONE’s publication criteria as it currently stands. Therefore, we invite you to submit a revised version of the manuscript that addresses the points raised during the review process.

We look forward to receiving your revised manuscript.

Kind regards,

Prof. Anat Gesser-Edelsburg, Ph.D.

Academic Editor

PLOS ONE

Journal Requirements:

Reviewers' comments:

Reviewer's Responses to Questions

**Comments to the Author**

1. If the authors have adequately addressed your comments raised in a previous round of review and you feel that this manuscript is now acceptable for publication, you may indicate that here to bypass the “Comments to the Author” section, enter your conflict of interest statement in the “Confidential to Editor” section, and submit your "Accept" recommendation.

Reviewer #1: All comments have been addressed

2. Is the manuscript technically sound, and do the data support the conclusions?

Reviewer #1: Yes

3. Has the statistical analysis been performed appropriately and rigorously? 

Reviewer #1: I Don't Know

4. Have the authors made all data underlying the findings in their manuscript fully available?

Reviewer #1: Yes

5. Is the manuscript presented in an intelligible fashion and written in standard English?

Reviewer #1: Yes

6. Review Comments to the Author

Reviewer #1: The authors have adequately addressed comments from the previous round of review of this manuscript.

It would even be better if the authors could briefly introduce the study design and also highlight ethical considerations about the study (this may be inserted in line 235). Other than that, this is a good paper to publish

7. PLOS authors have the option to publish the peer review history of their article (what does this mean?). If published, this will include your full peer review and any attached files.

Reviewer #1: No

---

## [Author Response · Author response to Decision Letter 1]

27 May 2021

We are delighted that our revisions in round 1 have met with approval and that only small adjustments are considered needed in this stage. Once again, we would like to thank the editor for the opportunity to revise our manuscript and the reviewer for their insightful suggestion.

Please find the response to the point raised by the reviewer below.

Reviewer #1: 

The authors have adequately addressed comments from the previous round of review of this manuscript.

It would even be better if the authors could briefly introduce the study design and also highlight ethical considerations about the study (this may be inserted in line 235). Other than that, this is a good paper to publish

RESPONSE: We agree that an introduction to the study design at the beginning of the ‘Methodology’ section would improve the coherence of that section. We have thus added an overview of the research design under line 235 as suggested by the reviewer, which also includes ethical considerations about the study.

---

## [Editor Report · Decision Letter 2]

31 May 2021

Lockdowns, lethality, and laissez-faire politics. Public discourses on political authorities in high-trust countries during the COVID-19 pandemic.

PONE-D-21-02692R2

Dear Dr. Perlstein,

We’re pleased to inform you that your manuscript has been judged scientifically suitable for publication and will be formally accepted for publication once it meets all outstanding technical requirements.

Kind regards,

Prof. Anat Gesser-Edelsburg, Ph.D.

Academic Editor

PLOS ONE
---

## [Editor Report · Acceptance letter]

4 Jun 2021

PONE-D-21-02692R2 

Lockdowns, lethality, and laissez-faire politics. Public discourses on political authorities in high-trust countries during the COVID-19 pandemic. 

Dear Dr. Perlstein:

I'm pleased to inform you that your manuscript has been deemed suitable for publication in PLOS ONE. Congratulations! Your manuscript is now with our production department. 

Kind regards, 

on behalf of

Prof. Anat Gesser-Edelsburg 

Academic Editor

PLOS ONE